# Is the Development of Hypo-Gammaglobulinemia Associated with Better Treatment Response in Patients with Rheumatoid Arthritis Using Rituximab?

**DOI:** 10.3390/jcm14196967

**Published:** 2025-10-01

**Authors:** Emine Gozde Aydemir Guloksuz, Serdar Sezer, Didem Sahin Eroglu, Sevgi Colak, Ayse Bahar Kelesoglu Dincer, Mucteba Enes Yayla, Emine Uslu, Mehmet Levent Yuksel, Recep Yilmaz, Elif Sinem Ates, Tahsin Murat Turgay, Gulay Kinikli, Askin Ates

**Affiliations:** 1Division of Rheumatology, Department of Internal Medicine, Ankara University Medical School, 06620 Ankara, Turkey; serdarsezer1987@hotmail.com (S.S.); dr.didemsa@gmail.com (D.S.E.); bahark@hotmail.com (A.B.K.D.); enesyayla@hotmail.com (M.E.Y.); drusluemine@gmail.com (E.U.); leventyuksel_52@hotmail.com (M.L.Y.); recep.yilmaz0621@gmail.com (R.Y.); tmturgay@hotmail.com (T.M.T.); gkinikli@gmail.com (G.K.); askinates1970@hotmail.com (A.A.); 2Division of Immunology and Allergy, Department of Internal Medicine, Ankara University Medical School, 06620 Ankara, Turkey; drsevgicolak@gmail.com; 3Department of Molecular Biology and Genetics, University of Tuebingen, 72074 Tubingen, Germany; elifsinemates@gmail.com

**Keywords:** rituximab, hypogammaglobulinemia, rheumatoid arthritis

## Abstract

**Objectives**: To determine the frequency of development of hypogammaglobulinemia in rheumatoid arthritis (RA) patients receiving rituximab (RTX) and to examine the relation between the development of hypogammaglobulinemia and RTX treatment response. **Methods**: The data of 165 RA patients who applied to our outpatient clinic between January 2010 and June 2021, and who received at least 2 courses of RTX with an interval of 6 months, were retrospectively evaluated. The demographic, clinical, and laboratory data, as well as treatment characteristics, were collected. **Results**: Of 165 patients, 35 (21.2%) developed hypogammaglobulinemia. In the multivariable analysis examining the risk factors for the development of hypogammaglobulinemia in RA patients receiving RTX, it was determined that having pre-treatment IgG value below 10.5 g/l (OR= 4.24 (95% CI 1.69–10.66) and the increase in the number of RTX courses (OR= 1.1 (95% CI 1.01–1.22) were independently associated risk factors. During their follow-up, patients who developed hypogammaglobulinemia and those who did not were compared. No difference was observed between DAS28-ESR levels, but CRP levels were significantly lower in the group that developed hypogammaglobulinemia. **Conclusions**: In this study, there was no difference in DAS28-ESR levels between patients with and without hypogammaglobulinemia, although a difference was observed in acute phase reactants, which are more objective parameters. This may be due to subjective parameters in DAS28-ESR scoring or other concomitant conditions such as fibromyalgia. Therefore, additional objective findings or methods may guide the evaluation of treatment response.

## 1. Introduction

Rheumatoid arthritis (RA) is a complex disease with a multifactorial pathogenesis, involving genetic, environmental, and immunological factors that contribute to its development and progression. Many cell populations, including B cells, several cytokines, and auto-antibodies, play a role in its pathophysiology [1]. Rituximab (RTX), an anti-CD20 monoclonal antibody, was developed for the treatment of lymphoproliferative diseases, especially for B-cell Non-Hodgkin Lymphoma, but is also frequently used in RA, Systemic Lupus Erythematosus (SLE), and anti-cytoplasmic antibody (ANCA) associated vasculitis [2,3,4,5,6].The use of RTX in the treatment of RA is effective both in controlling the symptoms and signs of the disease and in decreasing its long-term destructive effects by reducing radiographic progression [7].

Rituximab is a therapeutic agent that selectively targets the pre-plasma B cell population (such as immature B cells in the bone marrow, autoantigen-activated follicular B cells, and memory B cells) that express CD20 on their surface. Stem cells and long-lived CD20-negative plasma cells are not targets of RTX therapy. After binding of RTX to the cell surface receptor, targeted cells are killed by antibody-dependent cellular cytotoxicity, complement-dependent cytotoxicity, and/or phagocytosis and apoptosis in the reticuloendothelial system [8,9]. Removal of mature CD20-positive B lymphocytes, which are determined to differentiate into autoantibody-producing plasma cells, is considered the main effect of RTX. Therefore, it is effective in the treatment of autoimmune diseases in which the pathogenesis involves autoantibody formation [10]. Considering the close interaction between B and T cells, RTX has been found to be effective in RA, in which T cells play an important role in the pathogenesis [11].

The disadvantages of prolonged and recurrent B cell depletion with RTX therapy are suppression of protective antibodies, development of hypogammaglobulinemia, and a theoretical increased risk for infections [12,13]. In the literature, long-term hypogammaglobulinemia after RTX has been reported, especially in lymphomas treated with the combination of chemotherapy and/or stem cell transplantation [14,15,16]. There are few data on the long-term risk of the development of hypogammaglobulinemia in RA patients treated with RTX [13,17]. In addition, there is data available regarding the long-term safety profile of RTX in the treatment of RA [13,18]. Additionally, there is conflicting data in the literature showing the relation between the development of post-RTX hypogammaglobulinemia and serious infections [19,20,21]. However, only one study has investigated the relationship between hypogammaglobulinemia development after RTX treatment and treatment response in RA [22]. The aim of this study is to determine the frequency of development of hypogammaglobulinemia in RA patients receiving RTX and to examine the relationship between the development of hypogammaglobulinemia and RTX treatment response.

## 2. Materials and Methods

In this study, RA patients who applied to our outpatient clinic between January 2010 and June 2021, and who received at least 2 courses of RTX with an interval of 6 months, were included. All patients were aged ≥ 18 years and met the ACR/EULAR 2010 classification criteria for RA [23]. Patients who did not have at least 2 laboratory values examined at 6-month intervals, patients with missing baseline immunoglobulin data, and patients with known monoclonal gammopathy were excluded from the study. In our cohort, the indication for initiating rituximab (RTX) therapy was refractory disease. All patients had received at least three conventional synthetic DMARDs (csDMARDs) for ≥6 months before RTX initiation. RTX was prescribed to patients who failed to achieve an adequate response despite appropriate dosing and duration of csDMARD therapy and who demonstrated good adherence. Based on the treating physician’s clinical judgment, RTX was also considered in patients with inadequate response to other biologic DMARDs (particularly TNF inhibitors and/or JAK inhibitors) or in those deemed unsuitable for such therapies. Patients considered unsuitable included those with a history of malignancy, solid or hematologic pre-malignant lesions, or a family history of malignancy or autoimmune neurological diseases. These criteria were in line with national and international recommendations for RTX use in refractory RA, as well as the regulations of our national health insurance system. The standard regimen of 1000 mg RTX administered as two intravenous infusions 2 weeks apart was used in all patients. All patients were screened for hepatitis B virus (HBV), human immunodeficiency virus (HIV), and latent tuberculosis (TB) prior to RTX initiation, in accordance with national and international recommendations. Given the intermediate prevalence of TB and high prevalence of HBV in our country, PPD/IGRA tests were used for TB screening, and antiviral prophylaxis was initiated in patients requiring it. The demographic, clinical, and laboratory data, as well as treatment characteristics, were retrospectively collected. The concomitant immunosuppressive therapies administered during the period of RTX treatment were documented, as presented in Table 1. For patients who continued biologic drug therapy with RTX, acute phase reactant levels (erythrocyte sedimentation rate (ESR) and/or C-reactive protein (CRP)) along with immunoglobulin (Ig) levels before the first and 6 months after the last RTX courses were recorded. As for patients whose RTX treatment was changed due to secondary unresponsiveness, acute phase reactant levels and Ig levels before the first RTX course and up to 6 months after the last RTX dose were recorded. Hypogammaglobulinemia was defined when the value of one or more Ig types was under the limit of the normal laboratory reference range of our hospital (<7.51 g/L for IgG, <0.46 g/L for IgM, and <0.82 g/L for IgA). Low IgG level was considered mild if it was between 5 and 7.51, moderate if it was between 3 and 5, and severe if it was <3.

Disease Activity Score (DAS) 28-ESR (erythrocyte sedimentation rate) was used for determining the disease activity. DAS28-ESR at the time of initiation of the RTX therapy and at the time of check-up 6 months after RTX were defined as DAS28-ESR-1 and DAS28-ESR-2, respectively. All included patients had high disease activity (DAS28-ESR > 5.1) at baseline. RA disease activity was interpreted as remission (DAS28 -ESR< 2.6), low (2.6 < DAS28-ESR < 3.2), moderate (3.2 < DAS28-ESR < 5.1), or high (DAS28-ESR > 5.1).

The cases of serious infection, requiring hospitalization and/or IV antibiotic use, during RTX therapy, and comorbidities were also recorded.

The protocol of the study was approved by the Ankara University Ethics Committee with the number İ9-605-21/26 October 2021. This study was created in accordance with the 1964 Declaration of Helsinki and its later amendments.

### Statistical Analysis

Statistical analyses were performed using IBM SPSS Statistics for Windows, version 25.0 (IBM Corp., Armonk, NY, USA). The conformity of the variables to the normal distribution was examined by visual (histogram and probability graphs) and analytical methods (Kolmogorov–Smirnov/Shapiro–Wilk tests). Descriptive analysis median, 25th and 75th percentiles for non-normally distributed numerical variables; For ordinal and categorical variables, frequency tables are given. In comparisons between groups, Mann–Whitney U or Wilcoxon tests were used for numerical data that were not normally distributed, and Chi-square or Fisher tests were used for categorical variables. Roc analysis was performed to determine cutoff values predicting hypogammaglobulinemia. In the presence of significant cutoff values, positive likelihood ratios and Youden indices were calculated and the appropriate cutoff values were selected. In univariate analyses comparing patients with and without hypogammaglobulinemia, parameters with a *p* value of <0.25 were included in the multivariable analysis. However, since in some patients RF and CCP values were observed during the treatment period, and DAS28-ESH values corresponded to high disease activity in all patients, these variables were excluded from the analysis. For *p* < 0.05, the results were considered statistically significant.

## 3. Results

A total of 165 patients were included in the study. The median age of the patients was 56 (48–62.5), and 121 patients (73.3%) were female. At the start of RTX treatment, the median disease duration was 9 (4–15) years, and baseline DAS28-ESR was 5.5. The patients received a median of 7 (4–9) courses of RTX, and 29.1% (48 patients) had been previously treated with non-RTX biological agent(s). The mean follow-up duration of patients was 36 months (range: 12–60 months). The demographic and descriptive characteristics of our patients, their serological status and RF/anti-CCP titers, the DMARD treatments they were using, their comorbidities (including interstitial lung disease, pulmonary nodules, and other chronic diseases), and data on serious infections are summarized in Table 1. The concomitant immunosuppressive therapies administered during the period of RTX treatment were also documented, as presented in Table 1. During follow-up, 7 patients experienced infectious complications, ranging from mild upper respiratory tract infections to more severe conditions requiring hospitalization (e.g., pneumonia, cellulitis, renal tuberculosis, disseminated herpes zoster, and herpetic keratitis). No infection-related mortality occurred.

When the demographic data of patients with and without hypogammaglobulinemia were compared, there was no significant difference in age, gender, and disease duration before RTX treatment. Patients who developed hypogammaglobulinemia after RTX treatment had lower pre-treatment IgG and IgM levels and received a higher number of RTX doses (*p* < 0.001, *p* < 0.001, and *p* = 0.006, respectively). Other characteristics between groups are presented in Table 1.

A total of 35 patients (21.2%) developed hypogammaglobulinemia. The number of patients with low IgG, IgM, and IgA levels were 18 (10.9%), 20 (12.1%), and 9 (5.5%), respectively (Table 2).

After RTX treatment, patients with low IgG, IgM, or IgA had much lower basal globulin levels compared to those with normal globulin levels (*p* values < 0.001, <0.001, and 0.002, respectively). The median Ig levels before and after RTX treatment are summarized in Table 3.

Median immunoglobulin levels (IgG, IgM, and IgA) before and after RTX treatment in patients with and without hypogammaglobulinemia are shown in Figure 1. While significant reductions were observed in patients who developed hypogammaglobulinemia, no statistically significant changes occurred in those without hypogammaglobulinemia. The distribution and variability of immunoglobulin levels are further illustrated in Figure 2 using violin plots, highlighting the differences between the two groups

Median IgG, IgM, and IgA levels before and after RTX treatment in patients with and without hypogammaglobulinemia. Immunoglobulin (IgG, IgA, IgM) levels were compared between patients with and without hypogammaglobulinemia before and after rituximab (RTX) treatment. Between-group comparisons were performed using the Mann–Whitney U test or the Wilcoxon signed-rank test, as appropriate. Statistical significance was further assessed using logistic regression analysis (*p* < 0.05).

When ROC analysis was used to determine the basal IgG cut-off value in predicting the development of hypogammaglobulinemia, the ideal limit was calculated as 10.5; for this value, the sensitivity is 60% specificity, 76.2% positive likelihood ratio is 2.52, and the youden index is 0.36 (AUC: 0.726 95% CI 0.632–0.820 *p* ≤ 0.001).

In the multivariable analysis examining the risk factors for the development of hypogammaglobulinemia in RA patients receiving RTX, it was determined that having pre-treatment IgG value below 10.5 g/l (OR= 4.24 (95% CI 1.69–10.66) and an increased number of RTX doses (OR= 1.1 (95% CI 1.01–1.22) were independently associated risk factors (Table 4).

The results of the multivariable logistic regression analysis are presented in Figure 3. Among the examined factors, only low baseline IgG level (<10.5 g/L) was identified as an independent risk factor for post-treatment hypogammaglobulinemia.

Among patients who developed hypogammaglobulinemia during follow-up, DAS28-ESR levels and the proportion of moderate-to-high disease activity at the final course of RTX therapy were lower, although the differences were not statistically significant. CRP levels were significantly lower in the group that developed hypogammaglobulinemia (*p* = 0.034). In terms of immunoglobulin subtypes, a significant decrease in ESR was observed in patients with low IgG levels (*p* = 0.049), whereas this difference was not found in patients with low IgM levels (Table 5).

## 4. Discussion

Rituximab is a crucial agent in many different rheumatological conditions, especially in the treatment of RA. Although RTX does not directly target plasma cells, it may lead to a decrease in immunoglobulin levels in treated patients [24]. The incidence of hypogammaglobulinemia and associated infectious complications after RTX treatment is higher in malignancies than in non-malignant conditions [25]. In their study, Athni et al. described late-onset hypogammaglobulinemia and delayed neutropenia occurring after RTX, and emphasized that these complications are associated with an increased risk of infections [26]. Although the dose of RTX used in malignancies, particularly lymphomas, is not significantly higher than in rheumatoid arthritis (RA), the increased association of RTX with hypogammaglobulinemia and infections in cancer patients may be due to the additional impact of other chemotherapy agents. In addition, autoimmune diseases are more frequently associated with hypergammaglobulinemia [27]. Therefore, the development of hypogammaglobulinemia may be rarer in these patients.

In the study of Evangelatos et al., the development of hypogammaglobulinemia was observed in 36 (43.4%) of 83 RA patients [22]. Among these, the rate of patients with low IgM, IgG, and IgA was 31.3%, 24.1% and 7.2%, respectively. This ratio is higher than our study, which hypogammaglobulinemia was observed in 35 (21.2%) of 165 RA patients treated with RTX. As shown in some other studies, there is a negative relation between the number of RTX courses administered and the immunoglobulin levels [17,28,29]. In our study, the lower rate of development of hypogammaglobulinemia may be related to the shorter follow-up period and, in turn, the low number of RTX doses given. Both Evangelatos et al.’s and our study showed more IgM deficiency after RTX than other Ig isotypes [22]. Other studies have also demonstrated that the decrease in IgM levels after treatment in autoimmune diseases is greater than the decrease in IgG and IgA levels [13,28,30,31]. One reason may be that RTX preferentially depletes naive and unswitched B cells, both of which are precursors for IgM-producing cells [32]. Unswitched (IgM-bound) B cells are preferentially consumed by RTX in vitro, suggesting a low threshold and slow regeneration of these cells [33,34]. In addition, decreases in IgM levels after RTX could potentially be due to the presence of IgM in CD20-bearing plasmablasts that succumbed to RTX [35]. Serum IgG and IgA are mainly produced in the bone marrow by long-lived plasma cells that are not directly targeted by RTX [36]. The relatively higher rate of persistent low IgM following RTX may suggest a defect in B cell maturation into immunoglobulin-secreting cells in some of these patients. It may also be possible that RTX itself has an as-yet-undiscovered effect on B-cell ontogeny and may directly or indirectly affect IgM production and regulation [29].

In the study of Boleto et al., low initial globulin levels (<8 g/Lt) were shown as an independent risk factor for the development of hypogammaglobulinemia after RTX treatment, while concomitant MTX use was reported to be protective for the development of hypogammaglobulinemia [17]. In another study in the literature, there was a greater decrease in globulin levels after RTX in patients with low initial globulin levels [29]. It is known from other studies in the literature that some factors, including chronic corticosteroid use and advanced age, are associated with decreased gammaglobulin levels after RTX [13]. Nie et al. demonstrated that low baseline IgG is the most important predictor of hypogammaglobulinemia [37]. In their study, hypogammaglobulinemia was observed in 63.3% of autoimmune patients treated with RTX, with significant declines in IgG and IgM detected within the first three months of therapy. In addition, Opdam et al. conducted a large cross-sectional study including 322 RA patients treated with RTX and reported a 20% prevalence of hypogammaglobulinemia [38]. In the same study, severe cases (<4.0 g/L) were found to be rare. Higher cumulative RTX dose and age >65 years were identified as independent predictors, whereas concomitant MTX use demonstrated a protective effect. In our study, the results were compatible with these studies, and low baseline IgG was found to be an independent risk factor for the development of post-treatment hypogammaglobulinemia, but the protective effect of MTX use was not observed.

In our study, we investigated whether there is a relation between the development of hypogammaglobulinemia after RTX and the treatment response. As demonstrated for the first time in the study of Evangelatos et al., lower IgM levels after RTX were associated with better disease outcomes [22]. At the end of the follow-up period, patients with lower IgM levels had a lower DAS28-ESR score and more remission or lower disease activity than those with higher IgM levels. Similarly, in this study, DAS28-ESR levels were lower in patients who developed hypogammaglobulinemia after RTX; however, the difference did not reach statistical significance. In addition, CRP and ESR levels were lower in patients who developed hypogammaglobulinemia after treatment (*p* = 0.034 and *p* = 0.049, respectively). Although DAS28-ESR levels were found to be similar between patients with and without hypogammaglobulinemia, a difference was observed in acute phase reactants This may be due to a more subjective parameter, such as ‘patient global assessment’ in DAS28-ESR scoring. Concomitant fibromyalgia or non-inflammatory musculoskeletal symptoms may contribute to this condition. Therefore, the use of techniques such as ultrasound and magnetic resonance imaging (MRI), which provide additional objective findings, may guide the evaluation of treatment response. As reported in the study of Jensen Hansen et al., DAS28 score may not accurately reflect disease activity in some RA patients [39]. In addition, in our study, unlike Evangelatos’s study, no APR difference was observed in patients with low IgM after treatment, while an APR difference was found in patients with low IgG.

In the literature, there is a lack of data regarding the development of hypogammaglobulinemia during RTX therapy and response to RTX treatment. As far as we know, our study is the second study to examine the relationship between the development of hypogammaglobulinemia and disease activity after treatment. The major strengths of this study are its being a single center and the inclusion of a higher number of patients compared to other studies. However, our study also has some limitations. These include the retrospective design of the study, the short follow-up period, the small subgroup sizes and the lower number of RTX cycles received by our patients compared to other studies in the literature. Additionally, the low number of patients who developed hypogammaglobulinemia limits both the comparison between groups and the generalizability of the data. In this regard, more comprehensive and long-term prospective studies are needed.

In conclusion, our study demonstrated that hypogammaglobulinemia, particularly IgM deficiency, may occur in a considerable proportion of RA patients receiving rituximab therapy. Low baseline IgG levels were identified as an independent risk factor for the development of post-treatment hypogammaglobulinemia [17,22]. Furthermore, patients who developed hypogammaglobulinemia exhibited lower acute phase reactants, suggesting a potential association between immunoglobulin levels and treatment response [22,39]. However, DAS28-ESR scores did not differ significantly between patients with and without hypogammaglobulinemia. These findings emphasize the importance of close monitoring of immunoglobulin levels during rituximab therapy in RA. Nevertheless, larger and longer-term prospective studies are warranted to validate our results and clarify the clinical implications of hypogammaglobulinemia in this patient population.

## Figures and Tables

**Figure 1 jcm-14-06967-f001:**
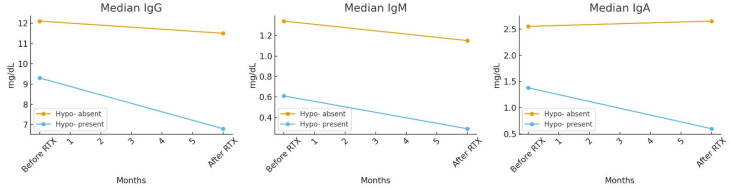
Median IgG, IgM, and IgA levels before and after RTX treatment in patients with and without hypogammaglobulinemia.

**Figure 2 jcm-14-06967-f002:**
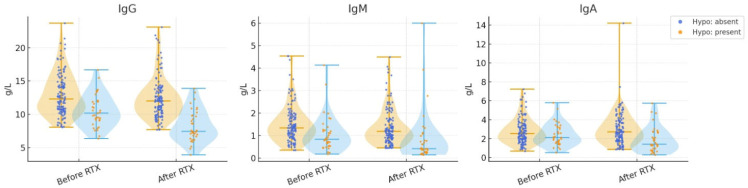
Distribution of IgG, IgM, and IgA levels before and after RTX treatment in patients with and without hypogammaglobulinemia (violin plots). Distribution of IgG, IgM, and IgA levels before and after rituximab (RTX) treatment in patients with and without hypogammaglobulinemia (violin plots). Sample sizes for each group are shown in the figure. Group comparisons were performed using the Mann–Whitney U or Wilcoxon signed-rank test, as appropriate.

**Figure 3 jcm-14-06967-f003:**
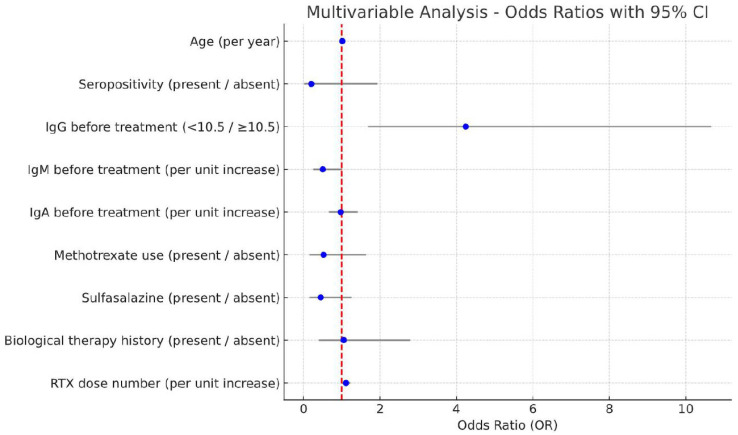
Multivariable logistic regression analysis of risk factors for post-treatment hypogammaglobulinemia. Risk factors associated with the development of hypogammaglobulinemia in patients receiving rituximab (RTX). Data are presented as odds ratios with 95% confidence intervals. Blue dots indicate the odds ratios for each variable, with horizontal lines representing the 95% confidence intervals. The red dashed vertical line represents the reference line at OR = 1.0.

**Table 1 jcm-14-06967-t001:** The demographic and descriptive characteristics of the patients.

	Total (*n* = 165)	Hypo-Gamma-Globulinemia Absent (*n* = 130)	Hypo-Gamma-Globulinemia Present (*n* = 35)	*p*
Age, years ^+^	56 (48–62.5)	55.8 (47.9–62.5)	56 (49–66.5)	0.430
Female gender *	121 (73.3)	94 (72.3)	27 (77.1)	0.566
Disease duration, years ^+^	9 (4–15)	9 (4–16.5)	10 (4.8–14)	0.590
RF titer, IU/mL ^+^	87 (28.5–224.5)	98.6 (39.2–279.9)	45.8 (11.3–150)	0.007
CCP titer, U/mL ^+¥^	127.6 (27.6–463.5)	158.1 (38.1–680.5)	37.8 (12.3–231.5)	0.002
IgG ^+§^	11.9 (10.3–14.2)	12.3 (10.7–14.8)	10.2 (9.1–12)	<0.001
IgM ^+§^	1.25 (0.83–1.8)	1.34 (0.94–1.85)	0.84 (0.54–1.27)	<0.001
IgA ^+§^	2.46 (1.81–3.39)	2.55 (1.91–3.42)	2.12 (1.55–3.21)	0.119
CRP, mg/L ^+§^	21.6 (10.1–43)	21.4 (9.9–45.3)	22.1 (10.5–38.1)	0.899
ESR, mm/hours ^+§^	34 (23–51.5)	34 (23–52.5)	35 (23–47)	0.660
DAS28-ESR ^+§^	5.5 (5.4–6)	5.5 (5.4–6)	5.8 (5.4–6.7)	0.203
Methotrexate *	39 (23.6)	34 (26.2)	5 (14.3)	0.142
Leflunomide *	65 (39.4)	52 (40)	13 (37.1)	0.759
Sulphasalazine *	58 (35.2)	50 (38.5)	8 (22.9)	0.086
Hidroxychloroquine *	64 (38.8)	52 (40)	12 (34.3)	0.538
GC *	115 (69.7)	93 (71.5)	22 (62.9)	0.321
Number of RTX cycles, median (IQR)	7 (4–9)	6 (3–9)	8 (5–13)	0.006
Serious infections, *n* (%)	6 (3.6)	3 (2.3)	3 (8.6)	0.110
Comorbidities, *n* (%)	37 (22.4)	30 (23.1)	7 (20)	0.698
Interstitial lung disease (ILD), *n* (%)	17 (10.3)	15 (11.5)	2 (5.7)	0.531
Pulmonary nodule, *n* (%)	12 (7.3)	10 (7.7)	2 (5.7)	1.000
Anti-HBc positivity, *n* (%)	37 (22.4)	30 (23.1)	7 (20)	0.698
Diabetes mellitus (DM), *n* (%)	28 (17)	22 (16.9)	6 (17.1)	0.975
Chronic kidney disease (CKD), *n* (%)	7 (4.2)	7 (5.4)	0 (0)	0.347
Atherosclerotic cardiovascular disease (ASCVD), *n* (%)	8 (4.8)	7 (5.4)	1 (2.9)	1.000

Values are given in numbers and percent * *n* (%); ^+^ median (Q1–Q3); ^§^ Rituximab levels before treatment, ^¥^ Anti-CCP antibody data were available for 149 patients (16 missing). ASCAD: atherosclerotic coronary artery disease, CCP: cyclic citrullinated peptide, CRF: chronic renal failure, CRP: C-reactive protein, DAS-28-ESR: disease activity score-28 erythrocyte sedimentation rate, DM: diabetes mellitus, ESR: erythrocyte sedimentation rate GC: Glucocorticoid, RF: rheumatoid factor, RTX: Rituximab.

**Table 2 jcm-14-06967-t002:** The change in immunoglobulin level after RTX application.

	n (%)
Hypo-gamma-globulinemia	35 (21.2)
Low IgG	18 (10.9)
Mild	16 (88.9)
Moderate	2 (11.1)
Severe	0 (0)
Low IgM	20 (12.1)
Low IgA	9 (5.5)

Values are given in numbers and percent n (%) RTX: Rituximab.

**Table 3 jcm-14-06967-t003:** Median Ig levels before and after RTX treatment.

	IgG g/L, Median (Q1–Q3)			IgM g/L, Median (Q1–Q3)			IgA g/L, Median (Q1–Q3)		
	Before Rtx	After Rtx	*p*	Before Rtx	After Rtx	*p*	Before Rtx	After Rtx	*p*
**Hypo-gamma-globulinemia absent**	12.1 (10.6–14.5)	11.5 (9.9–13.3)	<0.001 *	1.34 (0.94–1.86)	1.15 (0.76–1.67)	<0.001 *	2.55 (1.91–3.4)	2.65 (1.79–3.67)	0.975 *
**Hypo-gamma-globulinemia present**	9.3 (7.8–11.2)	6.8 (5.9–7.2)	<0.001 *	0.61 (0.46–0.87)	0.29 (0.21–0.41)	<0.001 *	1.38 (0.97–1.85)	0.6 (0.44–0.76)	0.008 *
** *p* **	<0.001 ^+^			<0.001 ^+^			0.002 ^+^		

* Wilcoxon test ^+^ Mann–Whitney-U test, Values are given in median (Q1–Q3), RTX: Rituximab.

**Table 4 jcm-14-06967-t004:** The risk factors for hypo-gamma-globulinemia.

	Univariate Analysis		Multivariable Analysis	
	OR (95% CI)	*p*	OR (95% CI)	*p*
Age, years (for every year)	1.02 (0.99–1.06)	0.240	1.01 (0.97–1.05)	0.610
Seropositivity (present → absent)	0.25 (0.03–1.84)	0.170	0.20 (0.02–1.93)	0.160
IgG before treatment (<10.5 → ≥10.5)	4.33 (1.97–9.49)	<0.001	4.24 (1.69–10.66)	0.002
IgM before treatment (for every unit increase)	0.43 (0.23–0.82)	0.010	0.50 (0.25–1.01)	0.054
IgA before treatment (for every unit increase)	0.78 (0.56–1.08)	0.130	0.97 (0.67–1.41)	0.870
Methotrexate use (present → absent)	0.47 (0.17–1.31)	0.150	0.52 (0.16–1.64)	0.270
Sulfasalazine (present → absent)	0.47 (0.20–1.13)	0.091	0.45 (0.16–1.25)	0.120
Biological therapy history (present → absent)	1.60 (0.73–3.53)	0.240	1.05 (0.40–2.79)	0.920
RTX dose number (for every unit increase)	1.12 (1.04–1.22)	0.004	1.11 (1.01–1.22)	0.036

RTX: rituximab.

**Table 5 jcm-14-06967-t005:** Comparison of disease activity between patients with and without hypo-gamma-globulinemia at the last course of RTX treatment.

	Hypo-Gamma-Globulinemia Absent (n = 130)	Hypo-Gamma-Globulinemia Present (n = 35)	*p*
DAS28-ESR ^+^	3 (2.3–5.1)	2.5 (2.1–3.4)	0.101
Disease activity *			0.163
Remission-low	80 (61.5)	26 (74.3)	
Moderate-high	50 (35.8)	9 (25.7)	
CRP ^+^	11.4 (5.1–24.1)	6.8 (3.4–14.3)	0.034
ESR ^+^	22 (12–37.5)	18 (9–26)	0.068
	**Normal IgG (n = 147)**	**Low IgG (n = 18)**	
DAS28-ESR ^+^	3 (2.2–4.1)	2.5 (2.1–3.4)	0.230
Disease activity *			0.454
Remission-low	93 (63.3)	13 (72.2)	
Moderate-high	54 (36.7)	5 (27.8)	
CRP ^+^	11.1 (4.8–22.7)	8.1 (3.3–13.6)	0.131
ESR ^+^	22 (12–35)	13 (6.5–29.5)	0.049
	**Normal IgM (n = 145)**	**Low IgM (n = 20)**	
DAS28-ESR ^+^	3 (2.2–4.3)	2.5 (2.1–3.7)	0.395
Disease activity *			0.567
Remission-low	92 (63.4)	14 (70)	
Moderate-high	53 (36.6)	6 (30)	
CRP ^+^	11.1 (4.9–22.7)	6.7 (3.6–14.3)	0.153
ESR ^+^	22 (12–35)	18.5 (9.5–24.5)	0.180
	**Normal IgA (n = 156)**	**Low IgA (n = 9)**	
DAS28-ESR ^+^	3 (2.2–4.1)	2.5 (2.1–4.2)	0.373
Disease activity *			0.575
Remission-low	99 (63.5)	7 (77.8)	
Moderate-high	57 (36.5)	2 (22.2)	
CRP ^+^	10.4 (4.8–21.9)	6.8 (1.9–19.3)	0.382
ESR ^+^	21.5 (11–35)	20 (13.5–35)	0.850

Values are given in numbers and percent * n (%); ^+^ median (Q1–Q3), CRP: C-reactive protein, DAS28: disease activity score 28, ESR: erythrocyte sedimentation rate, RTX: Rituximab.

## Data Availability

The data underlying this article will be shared on reasonable request to the corresponding author.

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
