# Peer review of "Is the Development of Hypo-Gammaglobulinemia Associated with Better Treatment Response in Patients with Rheumatoid Arthritis Using Rituximab?"

_jcm, 2025, doi:10.3390/jcm14196967_

Round 1
Reviewer 1 Report
Comments and Suggestions for Authors
It is an interesting, intriguing observation, but little bit too simplified.
There in no mention of severe neutropenia, cytopenia, immunologic distrubances such as low count of CD4+ T helper lymphocytes and concomitant virus infections.
Also you need to refresh literature with more recent papers.
Author Response
Response to Reviewer #1,
We would like to sincerely thank the reviewer #1 for the valuable comments and suggestions, which have greatly improved the quality of our manuscript.
Below are our point-by-point responses:
Reviewer #1 Comments and Responses
Comment 1:
“It is an interesting, intriguing observation, but a little bit too simplified. There is no mention of severe neutropenia, cytopenia, immunologic disturbances such as low count of CD4+ T helper lymphocytes and concomitant virus infections. Also you need to refresh literature with more recent papers.”
Response:
We sincerely thank the reviewer for this insightful and constructive comment. The primary objective of our study was to evaluate whether the presence of hypogammaglobulinemia influences the response to rituximab treatment. Therefore, factors potentially contributing to the development of hypogammaglobulinemia were not analyzed in depth within the scope of this work. Moreover, due to the retrospective design of the study, data on CD4+ T lymphocyte counts and concomitant viral infections were not available in the patients’ records and thus could not be incorporated into the analysis.
Nevertheless, in accordance with the reviewer’s valuable suggestion, we have revised the literature review and incorporated more recent publications to further strengthen both the background and the discussion sections of the manuscript.
We included:
- Reference [26]: Athni TS, Barmettler S. Hypogammaglobulinemia, late-onset neutropenia, and infections following rituximab. Annals of Allergy, Asthma & Immunology. 2023;130(6):699–712.
The publication cited as reference [26] has been mentioned in the Discussion section on page 8, lines 222-224 of the revised manuscript (marked version).
- Reference [37]: Nie Y, Zhang N, Li J, Wu D, Yang Y, Zhang L, et al. Hypogammaglobulinemia and Infection Events in Patients with Autoimmune Diseases Treated with Rituximab: 10 Years Real-Life Experience. J Clin Immunol. 2024 Nov;44(8):179.
The publication cited as reference [37] has been mentioned in the Discussion section on page 9, lines 251-253 of the revised manuscript (marked version).
- Reference [38]: Opdam MAA, Campisi LM, De Leijer JH, Ten Cate D, Den Broeder AA. Hypogammaglobulinemia in rheumatoid arthritis patients on rituximab: prevalence and risk factors. 2024 Jan 4;63(1):e1–2.
The publication cited as reference [38] has been mentioned in the Discussion section on page 9, lines 253-257 of the revised manuscript (marked version).
Please note that during the revision process, a technical issue occurred with the reference manager (Zotero), which unintentionally unlinked all in-text citations. As a result, all citations had to be reinserted, which may appear in the marked version as if the references were entirely modified. However, except for the newly added references required by the reviewers, no substantive changes have been made to the reference list.
We hope that these revisions adequately address the reviewer’s concerns and improve the clarity of our manuscript.
With kind regards,
Emine Gozde Aydemir Guloksuz
on behalf of all authors

Reviewer 2 Report
Comments and Suggestions for Authors
Rational
Your rational was to determine the frequency of development of hypogammaglobulinemia in RA patients receiving RTX and to examine the relationship between the development of hypogammaglobulinemia and RTX treatment response. However, the indications for using RTX in Refractory Rheumatoid Arthritis:
- Patient who have failed treatment with ≥ 3 synthetic disease-modifying anti-rheumatic drugs (sDMARDs) taken for ≥ 6 months.
- Patients have displayed good levels of adherence on failed regimens. You did not explain the causes of introducing RTX to the treatment protocol.
Methodology
What are the indications of RTX use? You mentioned that only48 patients had received previous biological therapy. The guidelines stated that RTX should be used in patients with treatment failure of one or more biological therapy.
Line 45: What was the dose of RTX?
Did you exclude HBV, HIV and TB in your patients before starting RTX? If yes, mention that in the methodology.
Line 64: All your included patients had high DAS score, you should mention that.
Statistical analysis
Line 79: Why was ROC curve performed for IgG only?
Line 80: The results of the ROC curve should be mentioned in the results section not in the section of the statistical analysis.
Results
In table (1): The mean DAS was higher in Hypogammaglobulinemia group, then how you state that, so there was no improvement in the outcome.
In table (1) you used DAS score, while in table (5) you used DAS-ESR, showing different results.
In figure (1) Ig measuring unit is mg/dL, while in methodology you used g/L. Use the same unit through the manuscript.
In figure (1) the x axis representing the treatment time should be mentioned in months.
The figures 1 and 2 were not mentioned in the text.
What is the explanation for the increase in IgA level after treatment in patients with no hypogammagloblineamia.
Line 105: Why was IgA not mentioned?
Line 106: You should mention the data of the table in sequence ie: table 1 then table 2.
The treatment guidelines for RA mentioned the combined use of methotrexate with RTX, in this study only 39 patients used methotrexate. What was your rationale?
Discussion
Line 201: DAS28 score may “be NOT cause” misleading in some RA patients.
Finally, there NO clear conclusion of the study.

Author Response
Response to Reviewer #2,
We would like to sincerely thank the reviewer #2 for the valuable comments and suggestions, which have greatly improved the quality of our manuscript.
Below are our point-by-point responses:
Reviewer #2 Comments and Responses
Comment 1:
Rational
Your rational was to determine the frequency of development of hypogammaglobulinemia in RA patients receiving RTX and to examine the relationship between the development of hypogammaglobulinemia and RTX treatment response. However, the indications for using RTX in Refractory Rheumatoid Arthritis:
- Patient who have failed treatment with ≥ 3 synthetic disease-modifying anti-rheumatic drugs (sDMARDs) taken for ≥ 6 months.
- Patients have displayed good levels of adherence on failed regimens. You did not explain the causes of introducing RTX to the treatment protocol.
Response 1 :
In our cohort, the indication for initiating RTX therapy in patients with rheumatoid arthritis was refractory disease. All patients had received at least three conventional synthetic DMARDs for ≥ 6 months prior to RTX initiation. RTX was prescribed to patients who failed to achieve an adequate response despite appropriate dosing and duration of csDMARD therapy and demonstrated good adherence to these treatment regimens. In addition, based on the treating physician’s clinical judgment, RTX was considered in patients who showed inadequate response to other biologic DMARDs (particularly TNF inhibitors and/or JAK inhibitors) or were deemed unsuitable for these therapies. Examples included patients with a history of malignancy, those with solid or hematologic pre-malignant lesions, or those with a family history of malignancy or autoimmune neurological diseases. These criteria were consistent with both national and international recommendations for the use of RTX in refractory RA, as well as with the regulations of our national health insurance system.
As suggested, we have explicitly stated in the Materials and Methods section the criteria for initiating RTX therapy in our cohort (page 2, lines 64-75).
Comment 2:
Methodology
What are the indications of RTX use? You mentioned that only 48 patients had received previous biological therapy. The guidelines stated that RTX should be used in patients with treatment failure of one or more biological therapy.
Line 45: What was the dose of RTX?
Did you exclude HBV, HIV and TB in your patients before starting RTX? If yes, mention that in the methodology.
Line 64: All your included patients had high DAS score, you should mention that.
Statistical analysis
Line 79: Why was ROC curve performed for IgG only?
Line 80: The results of the ROC curve should be mentioned in the results section not in the section of the statistical analysis.
Response 2:
Thank you for this important comment. We have now clarified the indications for RTX initiation in our cohort in the Methods section. RTX was prescribed to patients with refractory disease who had failed treatment with at least three csDMARDs (≥ 6 months each) with adequate adherence. In addition, RTX was considered for patients with inadequate response to or contraindications for biologic DMARDs (particularly TNF inhibitors and/or JAK inhibitors), in accordance with both national and international guidelines. This clarification has been added to the manuscript (page 2, lines 64-75).
“Line 45: What was the dose of RTX?”
We thank the reviewer for this observation. The standard regimen of 1000 mg RTX administered as two intravenous infusions 2 weeks apart was used in all patients. This information has been added to the Methods section (page 2, lines 75-76).
“Did you exclude HBV, HIV and TB in your patients before starting RTX? If yes, mention that in the methodology.”
We agree with the reviewer and appreciate this important point. All patients were screened for HBV, HIV, and latent TB prior to initiating RTX therapy. Our country is considered to have an intermediate prevalence of tuberculosis, and the frequency of latent TB is high. Therefore, all patients were screened for latent TB using PPD/IGRA in accordance with national and international recommendations. In addition, the prevalence of HBV is also high in our country; approximately one in three individuals over the age of 18 is either infected with HBV or has had a past HBV infection. Accordingly, HBV and HIV were screened prior to RTX therapy, and antiviral prophylaxis was initiated in all patients who required it. This has been clearly stated in the Methods section (page 2, lines 76-80).
‘’Line 64: All your included patients had high DAS score, you should mention that.”
We thank the reviewer for this suggestion. We have now indicated in the Methods section that all included patients had high DAS28 scores at the time of RTX initiation (page 3, lines 92-93).
“Statistical analysis — Line 79: Why was ROC curve performed for IgG only?”
We appreciate this question. ROC analysis was performed only for IgG because baseline IgG level was identified as an independent risk factor for post-treatment hypogammaglobulinemia in the multivariate analysis, whereas other immunoglobulins did not demonstrate significant associations.
“Line 80: The results of the ROC curve should be mentioned in the results section not in the section of the statistical analysis.”
Thank you for this valuable remark. We have moved the ROC curve results from the Statistical Analysis section to the Results section as recommended (page 6, lines 184-186).
Comment 3 :
Results
In table (1): The mean DAS was higher in Hypogammaglobulinemia group, then how you state that, so there was no improvement in the outcome.
In table (1) you used DAS score, while in table (5) you used DAS-ESR, showing different results.
In figure (1) Ig measuring unit is mg/dL, while in methodology you used g/L. Use the same unit through the manuscript.
In figure (1) the x axis representing the treatment time should be mentioned in months.
The figures 1 and 2 were not mentioned in the text.
What is the explanation for the increase in IgA level after treatment in patients with no hypogammagloblineamia.
Line 105: Why was IgA not mentioned?
Line 106: You should mention the data of the table in sequence ie: table 1 then table 2.
The treatment guidelines for RA mentioned the combined use of methotrexate with RTX, in this study only 39 patients used methotrexate. What was your rationale?
Response 3:
‘’In table (1): The mean DAS was higher in Hypogammaglobulinemia group, then how you state that, so there was no improvement in the outcome.’’
We thank the reviewer for this insightful comment. Table 1 presents the baseline characteristics of the patients. The median DAS28-ESR score was 5.8 (5.4–6.7) in the hypogammaglobulinemia group and 5.5 (5.4–6.0) in the non-hypogammaglobulinemia group, with no statistically significant difference between the two groups.
‘’In table (1) you used DAS score, while in table (5) you used DAS-ESR, showing different results.’’
We thank the reviewer for pointing this out. All DAS values reported in the manuscript refer to DAS28-ESR. The term “DAS” in Table 1 has been corrected to “DAS28-ESR” for consistency. In addition, all DAS28 expressions in the text and tables have been revised as DAS28-ESR. Table 1 presents the baseline DAS28-ESR values of the patients, whereas Table 5 shows the DAS28-ESR values after at least one course of RTX, which explains the observed differences.
‘’In figure (1) Ig measuring unit is mg/dL, while in methodology you used g/L. Use the same unit through the manuscript.’’
We agree with the reviewer. The units have now been standardized to g/L throughout the manuscript, including figures and tables.
‘’In figure (1) the x axis representing the treatment time should be mentioned in months.’’
Thank you for the suggestion. The x-axis of Figure 1 now clearly indicates treatment time in months. In addition, we have made some additional adjustments in Figure 1.
‘’The figures 1 and 2 were not mentioned in the text.’’
We thank the reviewer for this valuable observation. We have now cited Figures 1, 2 and 3 in the Results section. Specifically, Figure 1 presents the median immunoglobulin levels (IgG, IgM, and IgA) before and after RTX treatment in patients with and without hypogammaglobulinemia, while Figure 2 (added in line with Reviewer #4’s suggestion) illustrates the distribution and variability of these immunoglobulin levels using violin plots. Consequently, the figure that was previously numbered as Figure 2 has been renumbered as Figure 3 in the revised manuscript. These references have been added to the Results section.
‘’What is the explanation for the increase in IgA level after treatment in patients with no
hypogammagloblineamia.’’
We thank the reviewer for this question. There was a slight, but not statistically significant, increase in IgA levels in patients without hypogammaglobulinemia. We do not have a clear explanation for this minor, non-significant change.
‘’Line 105: Why was IgA not mentioned?’’
We thank the reviewer for this valuable comment. IgA has now been included in Table 5 in the revised manuscript (page 8).
‘’Line 106: You should mention the data of the table in sequence ie: table 1 then table 2.’’
We thank the reviewer for this helpful comment. The order of the tables has been revised accordingly, and the data are now mentioned sequentially in the revised manuscript.
‘’The treatment guidelines for RA mentioned the combined use of methotrexate with RTX, in this study only 39 patients used methotrexate. What was your rationale?’’
In our cohort, RTX was preferred according to individual patient characteristics and comorbidities. In some patients, contraindications or intolerance to methotrexate (e.g., liver disease, cytopenia, or gastrointestinal intolerance) limited its combined use. Due to patient preference, non-adherence to treatment, and physician choice, only 23.6% of our patients received concomitant MTX. As the reviewer has rightly pointed out, this proportion is relatively low compared with current treatment recommendations. However, since the rate of MTX use did not differ significantly between the main study groups, we believe that this factor did not have a primary impact on the study results. The potential effect of MTX use on the outcomes was further evaluated by including it in the multivariable analyses, and these results are presented in Table 4.
Comment 4:
Discussion
Line 201: DAS28 score may “be NOT cause” misleading in some RA patients.
Finally, there NO clear conclusion of the study.
Response 4:
‘’Line 201: DAS28 score may “be NOT cause” misleading in some RA patients.’’
We thank the reviewer for this helpful comment. The sentence has been revised for clarity as: “DAS28 score may not accurately reflect disease activity in some RA patients.” As suggested, this has been corrected in the Discussion section (page 9, lines 272-273).
‘’Finally, there NO clear conclusion of the study.’’
We thank the reviewer for this important comment. We would like to note that a Conclusion section was already included in the first revision of the manuscript.
The conclusion, as provided below, remains in the revised version (page 10, lines 285-293).
“In conclusion, our study demonstrated that hypogammaglobulinemia, particularly IgM deficiency, may occur in a considerable proportion of RA patients receiving rituximab therapy. Low baseline IgG levels were identified as an independent risk factor for the development of post-treatment hypogammaglobulinemia [17,22]. Furthermore, patients who developed hypogammaglobulinemia exhibited lower acute phase reactants, suggesting a potential association between immunoglobulin levels and treatment response [22,35]. These findings emphasize the importance of close monitoring of immunoglobulin levels during rituximab therapy in RA. Nevertheless, larger and longer-term prospective studies are warranted to validate our results and clarify the clinical implications of hypogammaglobulinemia in this patient population.’’
Please note that during the revision process, a technical issue occurred with the reference manager (Zotero), which unintentionally unlinked all in-text citations. As a result, all citations had to be reinserted, which may appear in the marked version as if the references were entirely modified. However, except for the newly added references required by the reviewers, no substantive changes have been made to the reference list.
We hope that these revisions adequately address the reviewer’s concerns and improve the clarity of our manuscript.
With kind regards,
Emine Gozde Aydemir Guloksuz
on behalf of all authors

Reviewer 3 Report
Comments and Suggestions for Authors
Thanks for the opportunity to review this paper, and thanks for your great efforts. Your study addresses a clinically important question regarding the immunological consequences of rituximab therapy in rheumatoid arthritis and their potential link to treatment response.
In general, the manuscript is well-structured, the statistical analysis is generally sound good, and the results are clearly presented.
However, some key aspects can be improved further.
Include All Referenced Figures: Figures 1 and 2 are mentioned but not present in the manuscript.
These are essential for understanding immunoglobulin trends and risk analysis.
just to ensure all figures are included, clearly labeled, and referenced appropriately inside the manuscript to have a better reflection.
Some of the methodological details are vague, such as the handling of missing data, the criteria for treatment, switching, and patient stratification need clarification. This would improve the transparency and reproducibility of the study.
Justify Statistical Decisions, the exclusion of RF and CCP from multivariate analysis is insufficiently explained. These biomarkers are highly relevant in RA disease activity and prognosis. Can you clarify the omission as it would strengthen the validity of the statistical model.
The conclusion suggests a link between hypogammaglobulinemia and better outcomes.
However, DAS28 scores showed no statistically significant difference. The conclusion should reflect this clearly and make it reasonable in the statement.
Expand the Limitations Section; it’s briefly acknowledged but needs more work.
Issues such as retrospective design, small subgroup size, and potential biases need emphasis.
A fuller discussion would help contextualize the results more appropriately.
In future revisions, I hope these points will improve the manuscript.
Author Response
Response to Reviewer #3,
We would like to sincerely thank the reviewer #3 for the valuable comments and suggestions, which have greatly improved the quality of our manuscript.
Below are our point-by-point responses:
Reviewer #3 Comments and Responses
Comment 1:
Include All Referenced Figures: Figures 1 and 2 are mentioned but not present in the manuscript.
These are essential for understanding immunoglobulin trends and risk analysis.
just to ensure all figures are included, clearly labeled, and referenced appropriately inside the manuscript to have a better reflection.
Response 1:
We thank the reviewer for this important comment. Figures 1, 2 and 3 have now been included in the revised manuscript. All figures are clearly labeled and appropriately referenced in the text to ensure proper understanding of immunoglobulin trends and risk analysis. We have now cited Figures 1, 2 and 3 in the Results section. Specifically, Figure 1 presents the median immunoglobulin levels (IgG, IgM, and IgA) before and after RTX treatment in patients with and without hypogammaglobulinemia, while Figure 2 (added in line with Reviewer #4’s suggestion) illustrates the distribution and variability of these immunoglobulin levels using violin plots. Consequently, the figure that was previously numbered as Figure 2 has been renumbered as Figure 3 in the revised manuscript.
Figure 1 appears in the Results section on page 5, lines 158-159.
Figure 2 appears in the Results section on page 5, lines 159-162.
Figure 3 appears in the Results section on page 7, lines 197-199.
Comment 2:
Some of the methodological details are vague, such as the handling of missing data, the criteria for treatment, switching, and patient stratification need clarification. This would improve the transparency and reproducibility of the study.
Response 2:
We thank the reviewer for this valuable comment. No statistical method was applied for imputing missing data. Anti-CCP antibody results were available for 149 patients, and this has now been indicated in Table 1 (page 4, lines 136-137).
For the other parameters evaluated in the study, no missing data were present.
In addition, the Methods section has been revised to clarify that patients with missing baseline immunoglobulin data were not included in the study (page 2, line 66).
Comment 3:
Justify Statistical Decisions, the exclusion of RF and CCP from multivariate analysis is insufficiently explained. These biomarkers are highly relevant in RA disease activity and prognosis. Can you clarify the omission as it would strengthen the validity of the statistical model.
Response 3:
We thank the reviewer for this valuable comment. Since the timing of RF and CCP measurements was not homogeneous across patients, seropositivity status was included in the multivariable analysis instead of the absolute levels (Table 4). However, in line with the reviewer’s suggestion, we have now additionally constructed a logistic regression model including RF and CCP levels, and the results are presented below.
Univariate analysis |
Multivariate analysis |
|||
|
OR (95% CI) |
p value |
OR (95% CI) |
p value |
Age, years (for every year) |
1,02 (0,99-1,06) |
0,24 |
1,02 (0,98-1,07) |
0,35 |
RF (for each unit increase) |
0,997 (0,994-1,00) |
0,026 |
0,998 (0,995-1,001) |
0,18 |
CCP (for every unit increase) |
0,999 (0,997-1,00) |
0,037 |
0,999 (0,998-1,00) |
0,18 |
IgG before treatment (<10,5 à ≥10,5) |
4,33 (1,97-9,49) |
<0,001 |
4,47 (1,64-12,13) |
0,003 |
IgM before treatment (for every unit increase) |
0,43 (0,23-0,82) |
0,010 |
0,63 (0,32-1,25) |
0,18 |
IgA before treatment (for every unit increase) |
0,78 (0,56-1,08) |
0,13 |
1,07 (0,72-1,61) |
0,73 |
Methotrexate use (present à absent) |
0,47 (0,17-1,31) |
0,15 |
0,75 (0,22-2,58) |
0,64 |
Sulfasalazine (present à absent) |
0,47 (0,20-1,13) |
0,091 |
0,44 (0,14-1,32) |
0,14 |
Biological therapy history (present à absent) |
1,60 (0,73-3,53) |
0,24 |
1,35 (0,49-3,74) |
0,56 |
RTX dose number (for every unit increase) |
1,12 (1,04-1,22) |
0,004 |
1,11 (1,00-1,22) |
0,051 |
Comment 4:
The conclusion suggests a link between hypogammaglobulinemia and better outcomes.
However, DAS28 scores showed no statistically significant difference. The conclusion should reflect this clearly and make it reasonable in the statement.
Response 4:
We thank the reviewer for this insightful comment. We have revised the Conclusion section to clearly reflect that, although hypogammaglobulinemia was observed in a considerable proportion of patients, DAS28 scores did not show a statistically significant difference. The text has been modified to ensure that the interpretation is consistent with the results and presented in a reasonable and accurate manner (page 10, lines 290-291).
Comment 5:
Expand the Limitations Section; it’s briefly acknowledged but needs more work.
Issues such as retrospective design, small subgroup size, and potential biases need emphasis.
A fuller discussion would help contextualize the results more appropriately.
Response 5:
We thank the reviewer for this valuable suggestion. We have expanded the Limitations section in the revised manuscript to discuss the retrospective design and small subgroup sizes more thoroughly. These additions provide a more comprehensive context for interpreting our results and enhance the transparency of the study (page 10, lines 281-282).
Please note that during the revision process, a technical issue occurred with the reference manager (Zotero), which unintentionally unlinked all in-text citations. As a result, all citations had to be reinserted, which may appear in the marked version as if the references were entirely modified. However, except for the newly added references required by the reviewers, no substantive changes have been made to the reference list.
We hope that these revisions adequately address the reviewer’s concerns and improve the clarity of our manuscript.
With kind regards,
Emine Gozde Aydemir Guloksuz
on behalf of all authors

Reviewer 4 Report
Comments and Suggestions for Authors
This study focuses on the development of hypogammaglobulinemia in patients with rheumatoid arthritis receiving rituximab. The topic is highly relevant, as the true prevalence of hypogammaglobulinemia among patients undergoing rituximab therapy remains unknown. Furthermore, there is insufficient data regarding the timing of its onset and the frequency of associated infectious complications. Additionally, clear clinical guidelines for the management of such patients are currently lacking.
Major recommendations:
- The authors should provide a more detailed description of the comparison groups, particularly regarding the medications used – whether rituximab was administered as monotherapy or in combination with other immunosuppressive agents. The manuscript should include information on the specifics of background therapy, prior use of biologic agents, and whether additional immunosuppressants were used in the comparison groups both before and after initiation of rituximab treatment.
- The temporal parameters must be clearly and precisely described: what was the duration of patient follow-up? How many cycles of rituximab therapy were administered? At what time point did hypogammaglobulinemia develop relative to the initiation of treatment?
- Information regarding concomitant and chronic comorbidities should be included. Did any infectious complications occur in patients who developed hypogammaglobulinemia? If so, were these complications more frequent compared to patients without hypogammaglobulinemia?
Minor recommendations:
- Figure 1 lacks metadata – information on statistical significance, sample sizes, and effect sizes is missing. It is recommended that this figure be redesigned, for example, using a box-and-whisker plot or a violin plot to better represent the distribution of data.
- Figure captions should be expanded to include brief descriptions of the presented results and the statistical methods used.
- The order of references in the text should be carefully checked. For instance, Table 3 is mentioned immediately after Table 1, which may confuse readers.
Author Response
Response to Reviewer #4,
We would like to sincerely thank the reviewer #4 for the valuable comments and suggestions, which have greatly improved the quality of our manuscript.
Below are our point-by-point responses:
Reviewer #4 Comments and Responses
Comment 1:
The authors should provide a more detailed description of the comparison groups, particularly regarding the medications used – whether rituximab was administered as monotherapy or in combination with other immunosuppressive agents. The manuscript should include information on the specifics of background therapy, prior use of biologic agents, and whether additional immunosuppressants were used in the comparison groups both before and after initiation of rituximab treatment.
Response 1 :
We sincerely thank the reviewer for this insightful comment. In accordance with the suggestion, we have substantially expanded the description of patient selection and background therapies in the Materials and Methods section. Specifically, we now clarify that in our cohort, the indication for initiating rituximab (RTX) therapy was refractory disease. All patients had received at least three conventional synthetic DMARDs (csDMARDs) for ≥6 months prior to RTX initiation. RTX was prescribed to patients who failed to achieve an adequate response despite appropriate dosing and duration of csDMARD therapy and who demonstrated good adherence. Based on the treating physician’s judgment, RTX was also considered in patients with inadequate response to biologic DMARDs (particularly TNF inhibitors and/or JAK inhibitors), or in those deemed unsuitable for such therapies. Details on screening procedures and treatment regimen have also been added.
Furthermore, we have included additional data on concomitant therapies in Table 1, which provides information on background medications used both before and after initiation of RTX. In addition, relevant details have been incorporated into both the Materials and Methods and Results sections. (page 2, lines 81-82) and (page 3, lines 123-124).
Comment 2:
The temporal parameters must be clearly and precisely described: what was the duration of patient follow-up? How many cycles of rituximab therapy were administered? At what time point did hypogammaglobulinemia develop relative to the initiation of treatment?
Response 2 :
Thank you for this valuable comment. We have now clarified the temporal parameters in the manuscript. The mean follow-up duration of patients was 36 months (range: 12–60 months). Patients received an average of 7 (4–9) cycles of rituximab therapy. These details have been added to the Results section (page 3, lines 118-120).
RTX was administered every 6 months, on days 0 and 15. This information has been added to the Materials and Methods section (page 2, lines 75-76). Immunoglobulin levels were assessed before each RTX infusion given at 6-month intervals, and the immunoglobulin levels presented in the Results section correspond to those measured immediately prior to the last RTX infusion each patient received.
Comment 3:
Information regarding concomitant and chronic comorbidities should be included. Did any infectious complications occur in patients who developed hypogammaglobulinemia? If so, were these complications more frequent compared to patients without hypogammaglobulinemia?
Response 3:
Thank you for this valuable comment. Patients’ comorbidities (including additional chronic diseases, interstitial lung disease, and the presence of pulmonary nodules) as well as data on serious infections have been added to Table 1, with corresponding details incorporated into the Results section (page 3, lines 121-122). During follow-up, 7 patients experienced infectious complications, ranging from mild upper respiratory tract infections to more severe conditions requiring hospitalization (e.g., pneumonia, cellulitis, renal tuberculosis, disseminated herpes zoster, and herpetic keratitis). No infection-related mortality occurred. This information has also been included in the Results section (page 3, lines 124-126).
Comment 4:
Figure 1 lacks metadata – information on statistical significance, sample sizes, and effect sizes is missing. It is recommended that this figure be redesigned, for example, using a box-and-whisker plot or a violin plot to better represent the distribution of data.
Response 4 :
We sincerely thank the reviewer for this constructive suggestion. In line with the recommendation, we have revised Figure 1 to include metadata regarding sample sizes, effect sizes, and statistical significance. Furthermore, to better illustrate the distribution of the data, Figure 1 has been redesigned using a violin plot, which is now presented as Figure 2 in the revised manuscript. This format was chosen as it more clearly demonstrates the spread and density of values across groups, as suggested by the reviewer.
Accordingly, the previous Figure 2 has been renumbered as Figure 3. We believe that these modifications improve the clarity and interpretability of the figures, and we thank the reviewer once again for this valuable feedback.
Comment 5:
Figure captions should be expanded to include brief descriptions of the presented results and the statistical methods used.
Response 5 :
We thank the reviewer for this suggestion. We have revised all figure captions to provide brief descriptions of the presented results and to include the statistical methods used. We believe these changes enhance clarity and allow readers to better interpret the figures.
We have updated Figure 1 as suggested. The caption now reads: ‘’Median IgG, IgM, and IgA levels before and after RTX treatment in patients with and without hypogammaglobulinemia. Immunoglobulin (IgG, IgA, IgM) levels were compared between patients with and without hypogammaglobulinemia before and after rituximab (RTX) treatment. Between-group comparisons were performed using the Mann–Whitney U test or the Wilcoxon signed-rank test, as appropriate. Statistical significance was further assessed using logistic regression analysis (p < 0.05).’’
We have updated Figure 3 as suggested. The caption now reads: “Risk factors associated with the development of hypogammaglobulinemia in patients receiving rituximab (RTX). Data are presented as odds ratios with 95% confidence intervals.”
In line with the reviewer’s recommendation, Figure 1 has been redesigned using a violin plot to better represent the distribution of data. This revised figure is now presented as Figure 2 in the manuscript.
Figure 2. Distribution of IgG, IgM, and IgA levels before and after rituximab (RTX) treatment in patients with and without hypogammaglobulinemia (violin plots). Sample sizes for each group are shown in the figure. Group comparisons were performed using the Mann–Whitney U or Wilcoxon signed-rank test, as appropriate.
Comment 6:
The order of references in the text should be carefully checked. For instance, Table 3 is mentioned immediately after Table 1, which may confuse readers.
Response 6:
Thank you for pointing this out. We have carefully reviewed the order of references and table citations in the manuscript. The mention of Table 3 immediately after Table 1 has been corrected to ensure a logical and sequential presentation of the tables, which we believe will improve clarity for the readers.
Please note that during the revision process, a technical issue occurred with the reference manager (Zotero), which unintentionally unlinked all in-text citations. As a result, all citations had to be reinserted, which may appear in the marked version as if the references were entirely modified. However, except for the newly added references required by the reviewers, no substantive changes have been made to the reference list.
We hope that these revisions adequately address the reviewer’s concerns and improve the clarity of our manuscript.
With kind regards,
Emine Gozde Aydemir Guloksuz
on behalf of all authors

Round 2
Reviewer 2 Report
Comments and Suggestions for Authors
Thank you for the careful and adequate responses, and for the changes made in the manuscript that made it more illustrative and clear.